# Access of older people to primary health care in low and middle-income countries: A systematic scoping review

**Saydeh Dableh**[1]*, **Kate Frazer**[1], **Diarmuid Stokes**[2], **Thilo Kroll**[1]

1 School of Nursing, Midwifery, and Health Systems, University College Dublin, Dublin, Ireland, 2 Library, University College Dublin, Dublin, Ireland

☯ These authors contributed equally to this work.

* Saydeh.dableh@ucdconnect.ie

**Data Availability Statement:** All relevant data are within the manuscript and its Supporting information files.

## Abstract

### Introduction

Ensuring access for older people to Primary Health Care (PHC) is vital to achieve universal health coverage, improve health outcomes, and health-system performance. However, older people living in Low-and Middle-Income Countries (LMICs) face barriers constraining their timely access to appropriate care. This review aims to summarize the nature and breadth of literature examining older people's experiences with access to PHC in LMICs, and access barriers and enablers.

### Methods

Guided by Arksey and O'Malley's framework, four databases [CINAHL, Cochrane, PubMed, and Embase] were systematically searched for all types of peer-reviewed articles published between 2002 and 2023, in any language but with English or French abstract. Gray literature presenting empirical data was also included by searching the United Nations, World Health Organization, and HelpAge websites. Data were independently screened and extracted.

### Results

Of 1165 identified records, 30 are included. Data were generated mostly in Brazil (50%) and through studies adopting quantitative designs (80%). Older people's experiences varied across countries and were shaped by several access barriers and enablers classified according to the Patient-Centered Access to Healthcare framework, featuring the characteristics of the care delivery system at the supply side and older people's attributes from the demand side. The review identifies that most access barriers and enablers pertain to the availability and accommodation dimension, followed by the appropriateness, affordability, acceptability, and approachability of services. Socio-economic level and need perception were the most reported characteristics that affected older people's access to PHC.

**Funding:** The authors received no specific funding for this work.

**Competing interests:** The authors have declared that no competing interests exist.

## Conclusions

Older people's experiences with PHC access varied according to local contexts, socioeconomic variables, and the provision of public or private health services. Results inform policymakers and PHC practitioners to generate policies and services that are evidence-based and responsive to older people's needs. Identified knowledge gaps highlight the need for research to further understand older people's access to PHC in different LMICs.

## Introduction

Universal Health Coverage (UHC) goals denote the right of all people to access healthcare services whenever needed, irrespective of their socioeconomic determinants of health. This is becoming increasingly challenging with global aging population trends [1]. While the right of all people to access health services remains universal as part of the 2020 Sustainable Development Goals, different plans exist at country level [2]. Despite an increase in the service coverage index from 2000 to 2019 showing advancements towards achieving UHC goals; 30% of people globally are unable to access basic health services [3]. The reasons for lack of access include the COVID-19 pandemic causing disruption of essential services in 92% of countries, and an increasing population of older people expected to reach 22% in 2050 [2, 3]. The risk is higher for 80% of older people, as they live in Low- and Middle-Income Countries (LMICs) [1] and struggle with access inequities aggravated because of strained healthcare resources and absent or inadequate social protection systems [4, 5]. Moreover, the risk of comorbidities increases in older age [6], often requiring comprehensive care. Enabling access to healthcare is the key to delivering integrated care [7] and preserving older people's right in attaining the highest possible standard of physical and mental wellbeing [8, 9].

Contextually, Primary Health Care (PHC) is identified as a strategy that can ensure UHC and the delivery of comprehensive care; However, PHC systems in LMICs often fail to meet UHC goals. PHC is defined as an approach to health aiming at attaining the optimal level of mental, physical, and social wellbeing through equitable distribution of essential and integrated care throughout the lifespan, for all people without hardship [2, 10]. It improves access, health outcomes, health equity, and health system efficiency especially during demographic and economic changes [11]. However, a scoping review on the performance of PHC systems in LMICs showed that PHC systems are still failing to achieve UHC [12]. It also highlighted the need for further research to improve PHC service delivery and address emerging challenges, especially in LMICs struggling with the double burden of non-communicable and communicable diseases [13].

Older people aged 60 years or over [5], face barriers to access affordable and comprehensive care. Access as a phenomenon is understudied in LMICs [1, 14] and has been evaluated by measuring utilization of health services [15, 16]. Access to health care has been inconsistently conceptualized in the literature; Levesque et al. (2013) define it as a journey that starts with perceiving the need for care, and continues with seeking, reaching, and utilizing care, which result in health consequences [17]. Health care access is affected by several factors pertaining to the supply and demand sides [15, 17]. Services need to be available, acceptable, affordable, approachable, and appropriate to the needs of the served population who should be able to perceive health needs, seek and reach care, pay for services, and engage actively in healthcare [17]. Access barriers or enablers can pertain to any of these factors. Carroll et al (2022) identified additional individual factors specific to older people like the capacity to make decisions,

rural/urban residency, and familial support. They highlighted that those factors vary among people aged between sixty and above, who should not be considered as a homogeneous group [1, 15]. Two integrative reviews summarizing the available literature on older people's access to PHC [16, 18], included studies from mainly Brazil, as LMIC [16, 18] and other developed countries [18]. They both suggest that older people face organizational, cultural, architectural, and geographical barriers constraining their timely access to appropriate PHC. Da Silva et al (2018) added to barriers the perception of health needs, the lack of confidence in public services, and the need to work in older age. In view of all these challenges, older people tend to seek care for acute and urgent conditions only, leading to poor health outcomes and increased health expenditures [19, 20]. Both reviews reported that homecare and telehealth facilitate access of older people to PHC. The reviews mentioned above, emphasized the importance of exploring older people's experiences with accessing PHC [16] which would contribute to the understanding of how social determinants of health influence access of older people to PHC across LMICs [21].

Evidence on factors determining older people's access to PHC has been generated mainly in high-income countries. While these integrative reviews, summarizing access of older people to PHC, are available, they are limited because they included studies published before 2015. One of them lacks focus on LMICs [18], the other included studies particularly published in Brazil, as a middle-income country [16]. New insight related to the same topic has been gained since the publication of these reviews. A scoping review makes it possible to summarize results from various sources in relation to what we currently know about older people's PHC access in different LMICs. This scoping review aims at mapping and summarizing the nature, features, and breadth of the available evidence related to older people's experiences with access to PHC in LMICs, and access barriers and enablers.

## Materials and methods

As stated in the published protocol [22], this review draws on Arksey and O'Malley's framework [23]. The Preferred Reporting Items for Systematic Reviews and Meta-Analyses—Extension for Scoping Reviews (PRISMA-ScR) guidelines were followed to ensure rigor in reporting (S1 Checklist).

### Review question

The presented data aims to answer the following two questions: 1) what are the experiences of older people with access to PHC in LMICs? 2) What are the PHC access barriers and enablers?

Summarized data answering the remaining sub-questions mentioned in the protocol [22] (conceptualization of access, PHC services, and interventions to maximize PHC access) are available in S1 File of the supporting information.

### Search strategy

We systematically searched four databases: PubMed, CINAHL, Cochrane library, and Embase, from 2002 up to November 2021 and the search was updated until May 2023. In revision to the review protocol [22], LILACS database was removed as the server could not process the long search strategy. Language filters were not applied and studies published in all languages but with English or French abstract were included. The search strategy combined key terms, index, and subject descriptors related to the four main concepts: 1) older people, 2) health service accessibility, 3) Primary health care, 4) LMICs. Use, utilization, or accessibility were other terms included to designate access, as authors might be using them interchangeably in the literature [17]. Same keywords were used across all databases whereas subject descriptors were

adapted for each database (see S1 Table for PubMed search strategy). Where available, Age filters (60 and above) were used. Finally, the reference lists of included studies were searched for possible additional references. For Grey literature, we searched the publication sections on official websites of the United Nations, World Health Organization, and HelpAge during December 2021 and May 2023. Only publications with empirical data were included.

## Study selection

After removing duplicates in EndNote X9, database search results were imported into Covidence software (Covidence systematic review software, Veritas Health Innovation, Melbourne, Australia, available at www.covidence.org). Two reviewers (SD and TK in 2021, SD and KF in 2023) independently assessed all retrieved titles, abstracts, and full texts against inclusion/exclusion criteria, detailed in Table 1. Conflicts were resolved by a third reviewer (KF in 2021 and TK in 2023) or discussed by all three reviewers to reach consensus.

## Charting and summarizing the data

Two reviewers (SD and TK) independently extracted 20% of the included studies. Then SD completed the data extraction after assessing the agreement level. Extracted data were exported from Covidence in an CSV file format document and saved in Excel sheet format (version 16.48) (S2 Table). In line with ScR methodology [24], quality appraisal for included studies was not conducted. Tabular and narrative synthesis were carried out after identifying and grouping similar concepts across studies to present data that fulfill the objectives of this review. Barriers and enablers mapped from included studies are classified according to the Patient-centered Access to Healthcare framework as presented by Levesque et al (2013).

**Table 1. Inclusion and exclusion criteria for study selection.**

| Inclusion | Rationale |
|---|---|
| Primary studies adopting any study design and all types of published reviews. | Mapping available published evidence and examining its nature, characteristics and volume. |
| Publications issued by UN, WHO, and HelpAge (reports including empirical data). | Mapping relevant gray literature issued by organizations that study, on the field, issues related to older adults. |
| Dating 2000 and above. | In 2002 two important events took place and led to the emergence of relevant research on aging: a) second UN assembly on aging that resulted in the Madrid International Plan of Action on Aging and b) Launching of the WHO policy framework on active aging. |
| Articles in all languages that include an abstract in English or French language. | Include as much as possible relevant studies from different LMICs that might not be published in English. |
| **Exclusion criteria** | **Rationale** |
| Studies conducted in high-income countries. | As mentioned above, the majority of older people reside in developing countries that are understudied. |
| Articles that do not include an abstract in English or French language. | For practicality purposes and time limitation, especially when screening titles and abstracts. |
| Grey literature other than publications issued by the UN, WHO, and HelpAge. Thesis, dissertations and conference posters will not be included. | For practical purposes and due to time limits. |
| Literature reviews and full articles that did not present the theme in the results. | To make sure that results answer to the proposed scoping review. |
| Studies with unavailable full text | Those are excluded only when full texts remain unavailable after contacting corresponding authors twice. |

WHO = World Health Organization; UN = United Nations; LMICs = low- and middle-income countries

## Results

Fig 1 summarizes the search and selection process resulting in the inclusion of thirty studies in this review.

### Characteristics of included studies

Table 2 summarizes the characteristics of included studies. It shows that 50% (n = 15) of studies are conducted in Brazil with seven published in Portuguese [25–31]. Almost half of the studies (53%) were published between 2019 and 2023 marking an increased interest in research focusing on older aged groups. Most studies (80%) adopted a quantitative design with the absence of participatory approaches. The role of participants in all studies was limited to invited membership of focus group interviews, individual interviewees, or respondents to surveys. Input from PHC service providers was sought in one study on care comprehensiveness [32].

### Participants' characteristics

Table 2 presents variation in sample sizes according to study designs. In most studies, samples consisted of older people, except for four studies examining services delivered to older people within PHCCs, and one study targeting families of older people. People from different genders, races, age groups, living areas, and socio-economic levels were represented with two studies targeting or advocating for the inclusion of older people with physical disabilities.

### Experiences of older people

The experiences of older people were specifically assessed in three included studies in this review [40, 43, 47]. Additional insight to older people's experiences was gained from studies depicting their satisfaction with delivered services [25] and their perception of service quality [33], and others analyzing the access and utilization of PHC services [28, 30, 36, 41, 42, 52, 53].

Older people's experiences with PHC access varied across countries and years. In Brazil [25], Saudi Arabia [42] and Myanmar [53] older people had positive experiences in general with PHC access despite their non-satisfaction with specific aspects like care organization and continuity. In South Africa [40, 43] older people reported negative experiences regarding the care they received from public facilities serving especially low-income areas, as care was not adapted to their needs. Older people's caregivers evaluated the performance of institutions and professionals as deficient when it comes to delivering care for older people [26]. Specific problems were the care discontinuity, shortage in medication and consumables supply, geographical and transportation barriers, delay in getting specialized services, and unpreparedness of care providers to deliver appropriate care for older people with complex health needs. This was supported by Martins et al. [30], highlighting that only a quarter of the services used by older people were comprehensive and oriented to their needs. However, a more recent study published in 2019 [25], showed that attributes of PHC were positively evaluated when it comes to care coordination, contact access, and comprehensiveness. A common fact noted across all studies is that older people and their families prefer to use private clinics instead of public services, whenever possible to avoid time-consuming, unreliable, and complex processes [25, 26, 40, 47, 52]. Experiences of older people with PHC were related to the type of health coverage and the living areas. In Brazil [41], fewest problems were reported by people having private plans, followed by those registered with the Family Health Strategy, then by people using traditional PHC clinics. In South Africa [40], services in low-income areas were less responsive to older people's needs compared to high-income areas. Available age-friendly policies were not

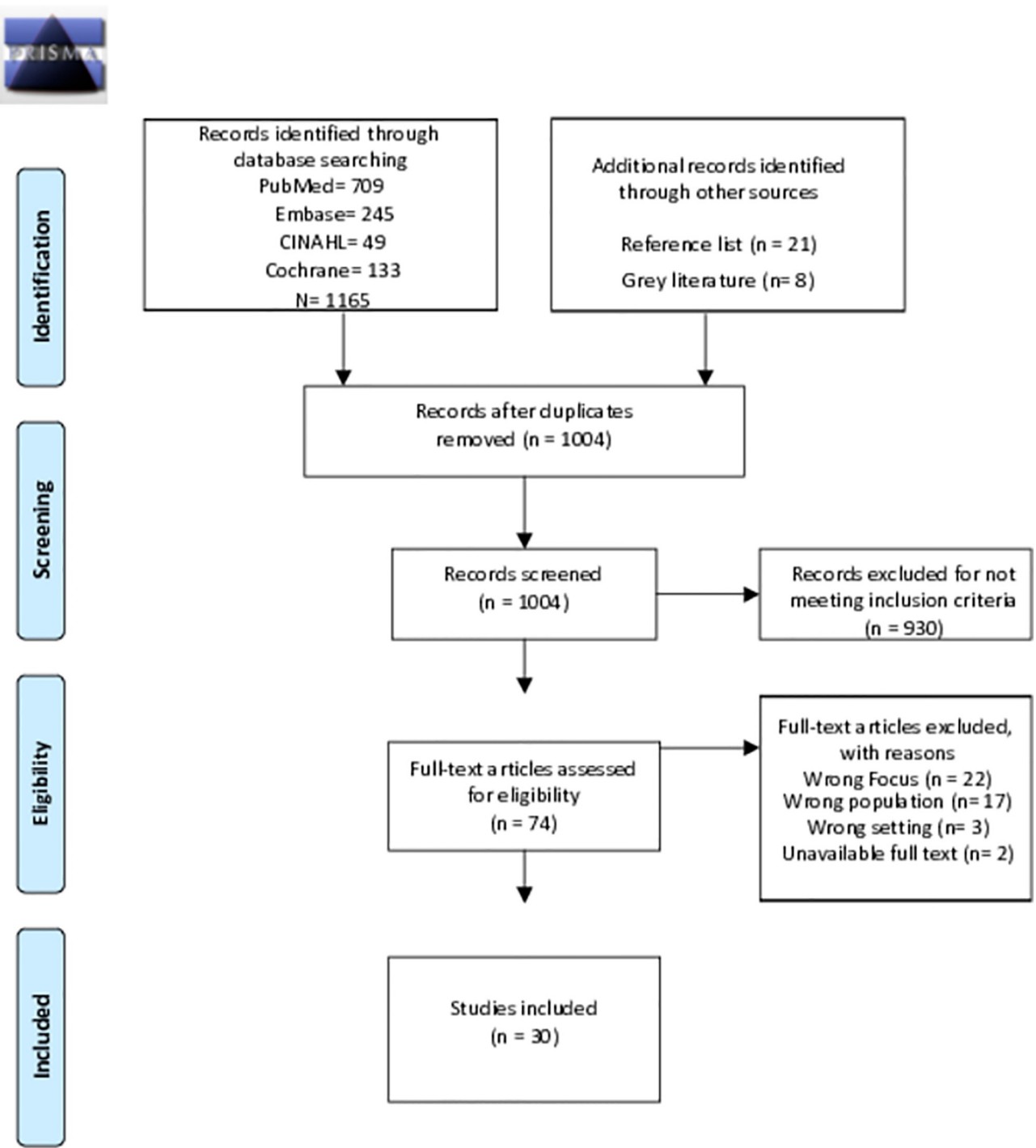

**Fig 1. PRISMA 2020 flow diagram.** From: Page MJ, McKenzie JE, Bossuyt PM, Boutron I, Hoffmann TC, Mulrow CD, et al. The PRISMA 2020 statement: an updated guideline for reporting systematic reviews. BMJ 2021;372:n71. doi: 10.1136/bmj.71. For more information, visit: http://www. prisma-statement.org/.

**Table 2. Characteristics of included studies.**

| Authors/ Year/ Country | Study Design | Aim | Context | Population Age & Number | Equity, Diversity, and Inclusion | Public & Patient Involvement |
|---|---|---|---|---|---|---|
| Amorim et al. (2020) Brazil [25] | Cross sectional | To assess the satisfaction of users of primary care services and to identify predictors of satisfaction to organize care and services (within municipalities that adhered to the PMAQ program) | Public primary healthcare centers, health post, outpost | People aged 60–75 (n = 18,671) | Men and women/ white and non-white skin/ literate and illiterate/ people aged above 75 not represented | None |
| Augusto et al. (2019) Brazil [33] | Cross sectional | To analyze factors associated with perception of PHC service quality by older people not having private health insurance. | Public primary healthcare centers, outpatient delivery without specifications | People aged 60–80 and above (n = 893) | Men and women/ white and non-white skin color/ Older people from all age ranges | None |
| Blay et al. (2008) Brazil [34] | Cross sectional | To determine whether older people have universal and egalitarian access to and appropriate use of health care services | Inpatient and outpatient (without further specification) | People aged 60–80 and above (n = 6,961) | Men and women/ white and non-white skin color/ Older people from all age ranges. The homeless/ institutionalized are not represented. | None |
| Bos (2007) Brazil [35] | Cross sectional | To assess whether the choice of health care provider and household income impact utilization and health of older people | Private clinics; Public clinics; Public primary healthcare centers; Outpatient care | People aged 60 to 69 (57% of sample) (n = 7,920) | Men and women/ white and non-white skin/ literate and illiterate/institutionalized not represented | None |
| Caner et al. (2019) Turkey [36] | Cross sectional | To examine health service utilization and satisfaction among older people | Private clinics; Public clinics; Public primary healthcare centers | People aged 65 and above (n = 3,170 first study in 2006 and 7,032 in 2015) (n = 737 second study in 2004 and 1,214 in 2014) | Men and women/different socioeconomic background/ literate and illiterate | None |
| Carreira et al. (2010) Brazil [26] | Qualitative | Identifying the difficulties experienced by families of older people having a chronic condition while seeking assistance at the Basic Health Unit (UBS) | Public primary healthcare centers; Homecare | 8 families (29 persons) | Dependant or independant, accessing either private or public clinics | None |
| Cesário et al. (2021) Brazil [27] | Cross sectional | To identify the conditions and trends in access and use of primary health care (PHC) services by Brazilian older adults considering the type of health service used and the difficulty in accessing the service. | Public primary healthcare centers | People aged 60–80 and above (n = 26,350 in 2008, n = 23,815 in 2013, n = 43,554 in 2019) | Older people from all ages, all races, men and women | None |
| Ferreira et al. (2020) Brazil [28] | Cross sectional | To characterize the access and use of health services among the older rural population from a municipality in southern Brazil and to identify factors associated with the choice of the Basic Family Health Unit (BFHU) as reference | Public primary healthcare centers | Age 60–80 and above (n = 1,030) | Older people from all ages, all races, men and women, literate and illiterate, residents of rural areas | None |
| Garcia-Ramirez et al. (2020) Colombia [37] | Cross sectional | To identify the determinants of healthcare utilization and to analyze the equality of healthcare utilization among elder Colombian patients | Inpatient and outpatient (without further specification) | People aged 60–80 and above (n = 23,694) | Older people from all ages, all socioeconomic levels, men and women, institutionalized not represented | None |

*(Continued)*

**Table 2.** (Continued)

| Authors/ Year/ Country | Study Design | Aim | Context | Population Age & Number | Equity, Diversity, and Inclusion | Public & Patient Involvement |
|---|---|---|---|---|---|---|
| Gao et al. (2022) LMICs [21] | Systematic review | To synthesize the available quantitative evidence on the relationship between socioeconomic inequalities and PHC utilization among older people | Community-based PHC centers, public and private clinics, first level hospitals, traditional healing clinics | Twenty studies presenting data on people aged 60 and above | Older people in general | None |
| Girondi et al. (2011) Brazil [29] | Integrative review | To identify studies on accessibility of older people with physical disabilities to primary health care services, from 1998 to 2008 | N/A | Scientific documents (n = 60) | Older people with physical disabilities | None |
| Gu et al. (2009) China [38] | Longitudinal (2002–2005) | To provide evidence on whether access to healthcare can increase healthy longevity at old ages | Inpatient and outpatient (without further specification) | People aged 60–80 and above (n = 15,972) | Older people from all ages, all socioeconomic and educational levels, men and women, rural and urban | None |
| Hu et al. (2019) China [39] | Cross sectional | To investigate older adults' choices of first-contact care when they felt ill in Zhejiang and Qinghai province, and the related potential pathways | Public primary healthcare centers | People aged 60 and above (n = 1,004) | Men and women, all socioeconomic and educational levels, dependent and independent | None |
| Kelly et al. (2019) South Africa [40] | Qualitative | To understand older persons' experiences of primary healthcare services in their communities | Private clinics, Public clinics Public primary healthcare centers | People aged 59–92 (n = 64) | Mostly women/all socioeconomic levels/ physically independent/ self-reporting of intact cognition. | None |
| Macinko et al. (2018) Brazil [41] | Longitudinal (cohort study 2015–2016) | To characterize healthcare access and utilization among older Brazilians | Private clinics; Public clinics; Public primary healthcare centers | People aged 50 years and older (mean age = 63) (n = 9,412) | Older people from different socioeconomic levels/ with and without functional limitation | None |
| Mahfouz et al. (2004) Saudi Arabia [42] | Cross sectional | To study the pattern of utilization of primary health care services and satisfaction among elderly people in Asir region | Public primary healthcare centers | People aged 60–85 (n = 253) | Rural and urban/men and women/ mostly illiterate | None |
| Martins et al. (2014) Brazil [30] | Review of documents Cross sectional | To establish a confrontation between theory and practice regarding the care focused on the health needs of older people in two districts of Porto Alegre | Public primary healthcare centers | People aged 60 years and older (mean age = 70) (n = 862) | Men and women, dependent people and those with cognitive limitations are not represented | None |
| Motsohi et al. (2020) South Africa [43] | Qualitative | To assess how older persons experience healthcare delivery at two primary healthcare clinics, and identify perceived gaps in health care to older people | Public primary healthcare centers | People aged 60–80 and above (n = 33) | Men and women, from different age groups | None |
| Nam et al. (2020) Korea [44] | Cross sectional | To evaluate the associations between payment exemption policies at the municipality level and the utilization of PHC services and the treatment rate of chronic diseases | Public primary healthcare centers | People aged 65–80 and over (n = 44,918) | Men and women, from different socioeconomic levels and age groups | None |

(*Continued*)

**Table 2.** (Continued)

| Authors/ Year/ Country | Study Design | Aim | Context | Population Age & Number | Equity, Diversity, and Inclusion | Public & Patient Involvement |
|---|---|---|---|---|---|---|
| Nwakasi et al. (2019) Ghana [45] | Cross sectional | To explore the association between lifestyle activities and outpatient care utilization rate. | Private clinics; Public clinics; Public primary healthcare centers; Outpatient care; Homecare | People aged over 60 (mean = 71.26) (n = 1,408) | Men and women/rural and urban/different socioeconomic and educational levels | None |
| Park et al. (2012) South Korea [46] | Cross sectional | To examine the extent to which equity in the use of physician services for the elderly has been achieved in Incheon | Outpatient without further specification | People aged 65–80 and over (n = 6,591) | Men and women, from different age groups | None |
| Paskulin et al. (2011) Brazil [31] | Cross sectional | To describe the use and geographic access of the elderly to primary healthcare (PHC) in Porto Alegre (RS), and to analyze the association between variables of interest to the study and access to PHC | Private clinics; Public clinics; Public primary healthcare centers; Homecare | People aged 60–80 and over (n = 292) | Men and women from all age groups and different socioeconomic and educational level | None |
| Peltzer et al. (2012) South Africa [47] | Cross sectional | To evaluate the degree of perceived responsiveness with outpatient and inpatient healthcare and to compare the experiences of individuals who used public and private healthcare services in South Africa. | Private clinics; Public clinics; Public primary healthcare centers; Outpatient care; Homecare | People aged 50 years and older (n = 3,840) | Older people in general | None |
| Placideli et al. (2020) Brazil [32] | Cross sectional | To evaluate the performance of comprehensive care for older adults in primary care services according to their managers and professionals and to analyze the relationship between performance and indicators of health planning and evaluation, in the state of SÃ£o Paulo, Brazil | Public clinics; Public primary healthcare centers; Homecare | Primary health care centers (n = 157) | N/A | None |
| Rodrigues et al. (2009) Brazil [48] | Cross sectional | To evaluate the utilization of healthcare services by elderly individuals who suffer from chronic diseases | Public clinics; Public primary healthcare centers; Homecare | People aged 65–80 and over (n = 2,889) | Men and women, having different skin colors, different age groups, and different educational and socioeconomic levels | None |
| Santos et al. (2020) Brazil [49] | Cross sectional | to identify architectural and communication barriers in Primary Health Care throughout Brazil (based on the results of the National Census of Primary Health Care Centers) | Public primary healthcare centers | Primary health care centers (n = 38,812) | Advocacy for inclusion and increased accessibility for older people and people with disability | None |
| Ssensamba et al. (2019) Uganda [50] | Cross sectional | To examine the readiness of public primary health care facilities to provide geriatric friendly services in Southern Central Uganda | Public primary healthcare centers | Public primary health care facilities (n = 18) | Advocacy for inclusion and increased accessibility for older people | None |

(*Continued*)

**Table 2.** (Continued)

| Authors/ Year/ Country | Study Design | Aim | Context | Population Age & Number | Equity, Diversity, and Inclusion | Public & Patient Involvement |
|---|---|---|---|---|---|---|
| Thumé et al. (2011) Brazil [51] | Cross sectional | To examine whether the FHS increases the utilization of home health care by the older people when compared with the Traditional PHC and to investigate the sources of the home health care used (whether public or private services) | Homecare | People aged 60–75 and over (n = 1,593) | Men and women, having different socioeconomic and educational levels, different skin color, having and not having functional limitations | None |
| Yam et al. (2019) Hong Kong [52] | Cross sectional | To assess changes over time in attitudes towards, and usage of, vouchers amongst older people in the community, and to assess the long term impact of the voucher scheme in encouraging the use of private primary care services | Private clinics; Public clinics; Public primary healthcare centers; Outpatient care | People aged 70–85 and over (n = 974 in 2016 1,026 in 2010) | Men and women | None |
| Sreerupa et al. (2017) Myanmar [53] | Mixed methods | To analyze the access to and utilization of healthcare services among older people and to identify barriers to access to healthcare (supply/ demand side) | Inpatient and outpatient care including private and public clinics, visiting nurses, midwives, drug stores | People aged 60 and above (n = 1000 for quantitative study, 20 to 25 focus groups joining 8–10 people each for the qualitative study) | Men and women, from all age groups, different socioeconomic levels, living in rural and urban areas, having or not functional and sensory limitations | None |

enough to avoid structural ageism and maladapted services that older people perceived as unreliable, strained, and difficult to access [43]. Additional factors shaped older people's experiences with PHC access; Women, those of advanced age, and those with higher education level evaluated their PHC access better. Those with chronic diseases, and greater use of services had more negative views, particularly concerning care orientation and coordination [25].

Factors leading to negative experiences are reported in the following section as barriers, whereas factors leading to positive experiences are reported as access enablers.

## Access barriers and enablers

In this review, the Patient-Centered Access to Health Care framework [17] was used to map the evidence from the 30 included studies. The process of access as defined by Levesque et al., (2013), has five dimensions representing the characteristics of the delivery system on the supply side: approachability, acceptability, availability and accommodation, affordability, and appropriateness. A set of abilities correspond to those dimensions representing the population's characteristics on the demand side: the ability to perceive, seek, reach, pay, and engage with healthcare (Fig 2). Factors hindering or facilitating access can relate to any of the dimensions or abilities, according to the context where access is happening.

**Approachability.** Older people who were not well informed about the available PHC services were less likely to access them [40].

**Acceptability.** Negative attitudes and behaviors of service providers [36, 40, 42, 43] that reflect structural ageism [43] are frequently reported as PHC access barriers followed by the lack of prioritization of older people who reach the facility to get services [26, 27, 41], and the low trust in nurses [40]. Older people who can choose their care-provider with whom they have a respectful relationship [53] are more likely to access PHC compared to people who

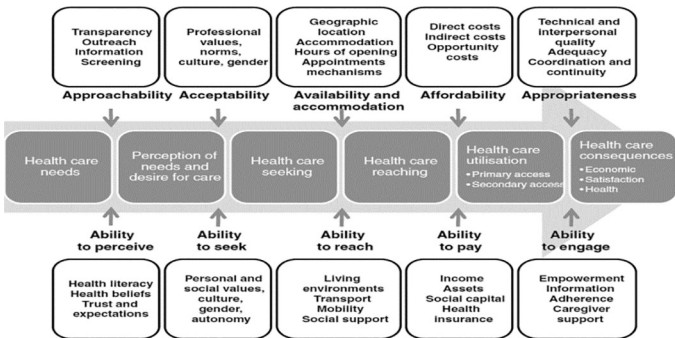

**Fig 2. Patient-Centered Access to Health Care framework [17].** Barriers represent factors either characterizing the healthcare delivery system or describing older people's attributes, reported in the included studies as constraining their use and access to PHC or leading to negative experiences. Whereas enablers represent such factors, reported as facilitating older people's access to PHC and leading to positive experiences. The evidence across the five dimensions is summarized in Table 3 and will be here described.

cannot make this choice [40, 43] or when interpersonal relationship with the provider is absent [26, 40].

**Affordability.**   The increased transportation cost [26, 35, 36, 43] and the lack of allocation of finances for geriatric care [50] leading to increased cost of services or increased contribution fees [36, 53] hinder PHC access. In contrast, free services [21, 31] and increased coverage rates [36] facilitate it.

**Availability and accommodation.**   As reported in included studies, the following factors can either facilitate or constrain the access of older people to PHC: availability of medical professionals especially specialists and geriatricians [26, 27, 29, 35, 36, 39–43, 49, 53]; distance and time to reach the nearest facility [25, 26, 28, 31, 36, 37, 39, 50, 53]; Availability of medication and other resources [25, 29, 39, 50, 53]; availability and adequacy of transportation [25, 26, 35, 36, 43]; availability of geriatric data to improve the service delivery [50]; mechanisms related to making appointments [26, 27, 29, 40–42]; architectural barriers [29, 30, 50]; scope of services especially the availability of home-based care [25, 30, 32, 47]; and service organization including opening hours [25], waiting time [27, 36, 40, 42, 47, 53], and waiting areas [29, 50, 53].

**Appropriateness.**   Identified factors that affect the responsiveness of PHC services to older people's needs, consequently their access to PHC, are: client-provider communication [41, 43, 47, 53], quality of the clinical examination [25, 36, 40, 41, 50]; training and skills in the provision of geriatric care [30, 32, 50]; involvement in the decision making [43, 47, 53]; care comprehensiveness [28, 30, 32, 50], continuity [26, 40, 41], and coordination [41]; delivery of structured patient education on geriatric care [26, 40, 42, 50]; geriatric assessment [50]; availability of geriatric guidelines [29, 50] and data [50].

**Ability to perceive.**   Literacy [29, 33, 34, 45, 46], income [28, 39, 48], need perception [28, 31, 33–35, 37, 44–46, 48, 51], and being used to a specific service [28] are all factors that affect the access of older people to PHC. Studies show that older age [33, 34, 44, 51], comorbidities and frailty [28, 31, 34, 35, 37, 46, 48, 51], and unhealthy lifestyles [45] increase the perception of need. Moreover, people with higher education [33, 34, 45, 46] are more likely to access PHC in general compared to people with lower educational level [29, 45]. However, those people will be more oriented toward services delivered by the private rather than those delivered by the public sector [31, 37, 39, 48].

**Ability to seek.**   Included studies show that PHC services, especially those offered by public providers, are mostly sought by people having a religion [46], low socioeconomic level [28,

**Table 3. Barriers and enablers to the access of older people to Primary Health Care in LMICs.**

| Supply Side | | Demand Side | |
|---|---|---|---|
| Characteristics of the delivery system | Barriers/Enablers | Older people's characteristics | Barriers/Enablers |
| **Approachability** | Information on available services [40] | **Ability to perceive** | Need [28, 31, 33–35, 37, 44–46, 48, 51]<br>Age [33, 34, 44, 51]<br>Literacy [21, 29, 33, 34, 45, 46]<br>Income [21, 28, 39, 48]<br>Use and preference [28] |
| **Acceptability** | Staff attitude and behavior [36, 40, 42, 43]<br>Care-provider relationship [26, 40]<br>Choice of the care provider [40, 43, 53]<br>Prioritization of older people [26, 27, 41]<br>Trust in nurses [40] | **Ability to seek** | Religion [46]<br>Race [27, 51]<br>Socioeconomic status (literacy/income/ health insurance) [21, 28, 31, 39, 44, 46, 48, 51]<br>Cognitive condition [46] |
| **Affordability** | Allocated finances for geriatric care [50]<br>Transportation cost [26, 35, 36, 43]<br>Service cost [21, 31, 36, 53]<br>Contribution fees/coverage rate [36, 53] | **Ability to pay** | Income [21, 29, 34, 37, 39, 45, 53]<br>Health insurance [21, 34, 37, 46] |
| **Availability & Accommodation** | Availability of medical professionals/ specialists/ geriatricians [26, 27, 29, 35, 36, 39–43, 49, 53]<br>Distance/time to the nearest facility [25, 26, 28, 31, 36, 37, 39, 50, 53]<br>Availability of medication and other resources [25, 26, 35, 36, 43]<br>Availability/adequacy of transportation [25, 26, 35, 36, 43]<br>Availability of geriatric data [50]<br>Mechanisms related to making appointments [26, 27, 29, 40–42]<br>Architectural barriers [29, 30, 50]<br>Scope of services/availability of home care [25, 30, 32, 47]<br>Service organization: opening hours [25]/ waiting time [27, 36, 40, 42, 47, 53]/ waiting areas [29, 50, 53] | **Ability to reach** | Gender [33, 34, 44, 46]<br>Age [28, 45, 46, 48]<br>Dwelling area [37, 45]<br>Literacy [21, 31, 39, 48]<br>Income [21, 34, 37, 39]<br>Mobility [39]<br>Community/familial support [34, 40, 46, 50]<br>Health insurance [21, 28, 31] |
| **Appropriateness** | Client-provider communication [41, 43, 47, 53]<br>Clinical examination quality [25, 36, 40, 41, 50]<br>Geriatric assessment [50]<br>Geriatric guidelines [29, 50]<br>Preparedness to provide geriatric care [30, 32, 50]<br>Involvement in the decision making [43, 47, 53]<br>Care comprehensiveness [28, 30, 32, 50]<br>Care continuity [26, 40, 41]<br>Care coordination [41]<br>Patient education on geriatric care [26, 40, 42, 50]<br>Geriatric data [50] | **Ability to engage** | Fear of physicians/examination [36]<br>Literacy [53] |

31, 44, 46, 51], minor race [27, 51], with absence of cognitive conditions [46]. People with high socio-economic levels seek care delivered through private facilities [28, 31, 39, 48].

**Ability to pay.** People having high income [21, 34, 37, 39] and private insurance [21, 34, 37, 46] are more likely to use PHC services compared to those having low income [29, 45] or living in poverty [53].

**Ability to reach.** Included studies indicate that women having time flexibility [33, 34, 44, 46], people with lower age [28, 46], higher income [21, 34, 37, 39], social support [34, 46], and better mobility [39] have enabled access to PHC. This also applies to people living in urban dwellings [37]. Having a high educational level [31, 39, 48] and private health insurance [28, 31] are barriers to accessing public PHC services. Older age [45, 46, 48] and lack of social support [40, 50] constrain people from reaching services especially if living in a rural area [45].

**Ability to engage.** Low educational level [33, 34, 45, 46] and fear of physicians and medical procedures [36] are noted as barriers to PHC access.

## Discussion

Results from this review summarize older people's experiences with access to PHC in LMICs along with access barriers and enablers.

### Experiences of older people

Results reporting experiences of older people with PHC access varied across countries. Older people in Brazil reported higher satisfaction with received care [25, 28], compared to people located in South Africa [40, 43]. The evidence in this review highlights inconsistencies and differs from previous reports published almost twenty years ago that indicated similar outcomes for older people living in developed and developing countries, regarding their access to PHC [9]. Varying experiences could be related either to the impact of implemented health system reforms on people's experiences, which is the case in Brazil [21, 38], or to the choice of study design. The qualitative research designs used in South Africa [40, 43] are better placed to explore the experiences of older people who tend to evaluate services positively despite existing challenges [53]. Notably, only 10% of the included studies adopted a qualitative design, marking a gap in exploring older people's lived experiences with PHC access.

People reported positive experiences when they felt heard [26, 40, 53], respected [52], and being involved in the decision-making regarding their own care plans [26, 40, 43, 47]. Negative experiences were associated with challenging factors reported under access barriers. Non-satisfaction with general practitioners' visits was also highlighted [41], reflecting their low preparedness in dealing with older people's complex needs [26]. Based on similar reported findings [9], the WHO identified training of healthcare professionals in clinical geriatrics, as one of the three main principles of age-friendly PHC. Moreover, people having chronic conditions and greater use of services were more likely to rate PHC services as poor especially in terms of care coordination and community and family orientation [25]. Within the same context, the WHO's report on integrated care of older people indicates that geriatric care and management of chronic conditions are not emphasized in training curricula, although older people are the most frequent users of healthcare services [54]. The report shines a light on the fragmented care and the lack of coordination between care levels, multiple facilities, and professionals, impeding the management of older people's complex issues [54]. Several included studies reported that instead of using public services, older people and their families prefer to seek care delivered through the private sector [26, 28, 41] whenever possible, seeking an easier access [26, 41, 43, 47] and a better quality of services [26, 28, 30, 40, 41, 43, 52].

### Access barriers and enablers

Categorizing access barriers and enablers according to the Patient-Centered Access to Healthcare framework was useful to highlight care delivery dimensions and characteristics of older people that shape their experiences with access to PHC.

To analyze factors pertaining to access dimensions from the supply side, the number of factors per dimension and the frequency of their reporting in included studies were considered. Findings show that most access barriers and enablers pertain to the availability and accommodation dimension, followed by the appropriateness, affordability, acceptability, and then approachability of services. Although described factors match with those reported previously in the literature [9, 18], findings are partially consistent with the work of Khanassov et al. (2016) who stated that approachability, availability, and affordability are the most evaluated access dimensions and that acceptability and appropriateness of care need further examination [20]. Being informed of available services is a factor that was reported in one study [40] to denote approachability, a dimension that marks the beginning of the healthcare access journey

(Fig 2); older people cannot access PHC services if they do not access information about their availability and use.

*Acceptability* of PHC services refers notably to staff attitudes, and behaviors [36, 40, 42, 43] that shape the client-provider relationship [26, 40, 53], ability to choose the preferred provider [40, 43, 53], and prioritization of older people's needs when they access PHC facilities to get services [26, 27, 41].

Within the *affordability* dimension, high cost of transportation [26, 35, 36, 43] and services [36, 53], along with high contribution fees [36, 53] were recognized as barriers to PHC access.

As for the *appropriateness* dimension, included factors described the adequacy of provided services against older people's needs. The clinical examination quality [25, 36, 40, 41, 43, 47, 50, 53] including time dedicated to every client, thorough geriatric assessment, explanation of the disease and related care plan, and shared decision making, was commonly reported. The client-provider communication [41, 43, 47, 53] entailing listening skills, respect, and privacy was also emphasized. A gap resulting from the complex needs prevailing in older age and the low preparedness of professionals to deliver geriatric [26, 30, 32, 40, 42, 50], comprehensive [28, 30, 32, 50], and continuous [26, 40, 41] care was also highlighted.

Factors pertaining to the *availability and accommodation* dimension were numerous and reported in several included studies. Many articles emphasized issues related to the availability of physicians and specialist [26, 27, 29, 35, 36, 39–43, 49, 53] affecting the waiting time at the facilities [27, 36, 40, 42, 47, 53]; the availability of adequate transportation means [25, 26, 35, 36, 43]; the availability of medical resources [25, 29, 39, 50, 53]; the availability of diverse services [25, 30, 32, 47]; difficulties in making appointments [26, 27, 29, 40–42]; and time and distance required to reach the nearest facility [25, 26, 28, 31, 36, 37, 39, 50, 53]. However, studies that considered the architectural barriers were scarce [29, 30, 50], especially considering that this could be a challenging aspect constraining the access of older people to PHC. A meta-synthesis, summarizing the barriers constraining the access of older people with disabilities to PHC in LMICs, reported that in addition to the logistical and physical environmental factors, informational barriers from both demand and supply sides and a set of attitudes and cultural beliefs related specifically to disability were identified [55].

Regarding the demand side, the most reported characteristics of older people that affected their access to PHC were their socio-economic level and their need perception. Higher income, improved literacy, and availability of private insurance were correlated to enabled PHC access in some studies [21, 33, 34, 37, 39, 45, 46] and to a constrained access in others [28, 31, 39, 48]; when studies focused specifically on the access to, or use of public services, a favorable socio-economic level was identified as an access barrier, because wealthy and educated older people tend to seek and reach services delivered by the private sector. The inconsistency in the association between socioeconomic inequalities and utilization of PHC by older people in LMICs was also reported by Gao et al. (2022). They suggested that the increased use of PHC by people with low socioeconomic status is an exception, which is only relevant in contexts where governments implemented several strategies to promote equitable access to PHC, like in Brazil, Cuba, Taiwan, and China [21]. Older age is a factor that increases the perception of need [33, 34, 44, 51]; however, it negatively affects the ability to reach PHC [45, 46, 48] especially in the presence of functional [48] and cognitive limitations [46]. Having a social network [34, 40, 46, 50] is linked to a better ability to reach services and a lower need perception due to better health outcomes [45]. Living in rural areas negatively affects the ability of older people to reach PHC services [37, 45]. Due to occupational and time flexibility, females seem to have an enabled access to PHC services [25, 34, 44, 46] compared to males [36].

## Access to PHC: A global challenge

In line with available literature [18, 56], results show that PHC systems in all countries are still falling short from meeting older people's needs, with different levels of effective performance across countries. Access problems exist even in high-income countries with common barriers and enablers being reported [57, 58]. However, high-income countries are more agile in terms of health system reforms and policy generation to meet the demographic transition implications, with focus on vulnerable groups' needs [59–61].

Enabled access to PHC is fundamental because it is associated with healthy longevity at older age [38] care continuity and coordination [33], along with improved health equity, and health system performance [11]. A systematic review of studies generated mainly in high-income countries has shown that hospital admissions for ambulatory care sensitive conditions decreased with enhanced accessibility to PHC [62]. Tailoring PHC services to be more responsive to older people's needs is possible. A recently published study from Sweden [59], exploring the experiences of older patients with a tailored PHC unit reported positive results. Older people felt as being in safe hands when having easy access, welcoming and skillful providers in geriatric care, and being treated in calm environments through patient-centered approach. Mirza et al. (2022) state in their scoping review that the perspectives of older people are not sought when restructuring PHC services, especially in rural areas. They suggest integrating PHC users' views on healthcare delivery models through co-designing approaches [63].

## Strengths and limitations

The key strength of this review is the robust conduct of the review process [22] which increases the process traceability and replicability. However, we acknowledge that some relevant studies could be missed due to limiting search years and inclusion of four databases. To balance those limitations, language filters were not used, and Cochrane LMICs filters were adopted to adhere to the utmost possible quality standards.

## Implications for practice and policy making

The review provides explicit understanding of access to PHC in LMICs which is essential for the growing older population. Results intel policy makers and inform PHC service providers to generate policies and services that are evidence-based and responsive to older people's needs. An exploration of older people's experiences with PHC access is needed, at a national level, to inform reforms of health systems. The presented evidence highlight dimensions that should be addressed or further explored while taking into consideration older people's abilities.

## Implications for research

On the conceptual level, this review adds to the guiding framework of Levesque et al., (2013) sub-dimensions that are most relevant to the access of older people to PHC in LMICs' context (Fig 3). The review also highlights research gaps and areas that should be further explored. Quantitative data should be complemented with evidence generated by qualitative designs that are suitable to explore older people's experiences. Older people with lived experiences, their families and care providers should be actively involved in research processes through participatory designs. Exploring the PHC access in different LMICs and including older people with cognitive and functional limitations are needed. Different access dimensions and population abilities need further exploration like the acceptability dimension and the ability of older people to engage with care.

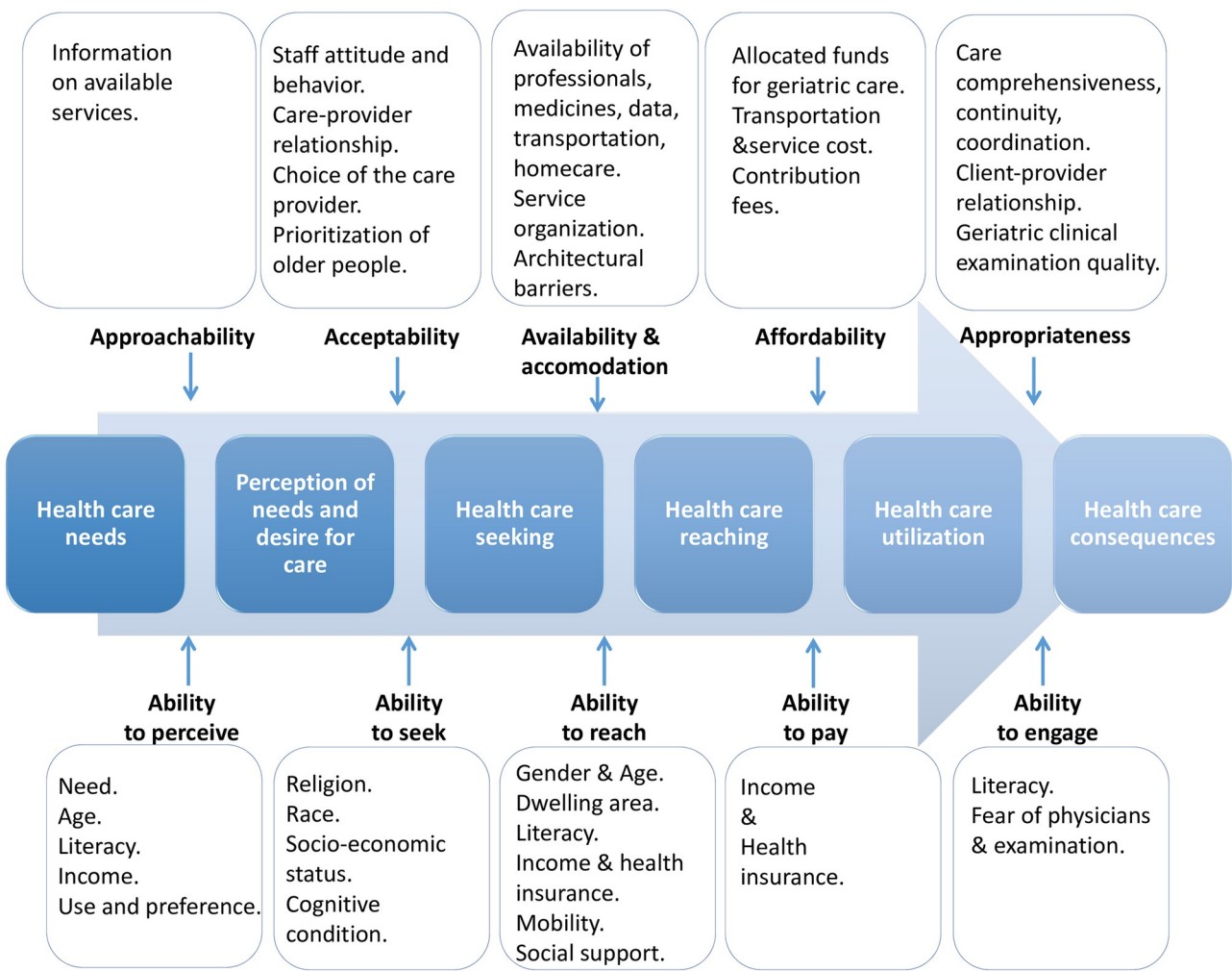

**Fig 3. Conceptual framework for the access of older people to PHC in LMICs.** Adapted from the Patient-Centered Access to Health Care framework [17].

## Conclusions

Granting access to PHC is essential to achieve UHC and healthy aging. However, results of this scoping review revealed that older people in LMICs are still unable to access adapted and integrated care. Older people's experiences with PHC varied across countries and were influenced by local context of care provision, the socioeconomic level of living areas, and the provided type of health coverage. Experiences were shaped by several access barriers and enablers that pertain either to supply or to personal characteristics of older people seeking care. Knowledge gaps of older people's access to PHC were also identified, highlighting a need for research to understand further the challenges faced by older people and to address them.

## Supporting information

**S1 Checklist. PRISMA-ScR checklist.**
(PDF)

**S1 File. Additional reported data.**
(DOCX)

**S1 Table. Search strategy PubMed.**
(DOCX)

**S2 Table. Full data set.**
(XLSX)

## Author Contributions

**Conceptualization:** Saydeh Dableh, Kate Frazer, Thilo Kroll.

**Data curation:** Saydeh Dableh, Kate Frazer, Thilo Kroll.

**Formal analysis:** Saydeh Dableh, Kate Frazer, Thilo Kroll.

**Investigation:** Saydeh Dableh, Kate Frazer, Thilo Kroll.

**Methodology:** Saydeh Dableh, Kate Frazer, Diarmuid Stokes, Thilo Kroll.

**Project administration:** Saydeh Dableh, Kate Frazer, Thilo Kroll.

**Supervision:** Kate Frazer, Thilo Kroll.

**Validation:** Kate Frazer, Diarmuid Stokes, Thilo Kroll.

**Writing – original draft:** Saydeh Dableh.

**Writing – review & editing:** Saydeh Dableh, Kate Frazer, Diarmuid Stokes, Thilo Kroll.

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
