## [Decision Letter · Decision Letter 0]

15 Nov 2023

PONE-D-23-19442Access of older people to primary health care in low and middle-income countries: a systematic scoping reviewPLOS ONE

Dear Dr. Saydeh Dableh,

Thank you for submitting your manuscript to PLOS ONE. After careful consideration, we feel that it has merit but does not fully meet PLOS ONE’s publication criteria as it currently stands. Therefore, we invite you to submit a revised version of the manuscript that addresses the points raised during the review process.

We look forward to receiving your revised manuscript.

Kind regards,

Ada Aghaji

Academic Editor

PLOS ONE

Journal Requirements:

2. Our internal editors have looked over your manuscript and determined that it is within the scope of our Aging in Human Health and Disease Call for Papers. This call for papers aims to highlight the excellent work being done by researchers across the world on the subject of aging. Additional information can be found on our announcement page: https://collections.plos.org/call-for-papers/aging-in-human-health-and-disease/. If accepted, your submission will be included within the collection. Please note that being considered for the Collection does not require an additional peer review beyond the journal’s standard process and will not delay the publication of your manuscript if it is accepted by PLOS ONE. If you have any questions or concerns about this process, please contact the journal at plosone@plos.org

Additional Editor Comments:

The authors should be congratulated for researching into this important topic. In addition to the comments made by the reviewers, the following should be addressed.

1. For the search strategy for PubMed please include the number of articles at each stage.

2. In the PRISMA flow diagram, please give reasons for why 930 articles were excluded.

3. A protocol paper has been published and referenced, and you mentioned changes to the protocol in the letter to the editor. These should be reflected in the methods.

Reviewers' comments:

Reviewer's Responses to Questions

**Comments to the Author**

1. Is the manuscript technically sound, and do the data support the conclusions?

Reviewer #1: Yes

Reviewer #2: Yes

2. Has the statistical analysis been performed appropriately and rigorously? 

Reviewer #1: Yes

Reviewer #2: Yes

3. Have the authors made all data underlying the findings in their manuscript fully available?

Reviewer #1: Yes

Reviewer #2: Yes

4. Is the manuscript presented in an intelligible fashion and written in standard English?

Reviewer #1: Yes

Reviewer #2: Yes

5. Review Comments to the Author

Reviewer #1: PLOS ONE

Title :- Access of older people to primary health care in low and middle-income countries: a systematic scoping review –

1. The adequacy of the design and thoroughness in the execution of the research:-

The topic addresses the provision of access of older persons to Primary Health Care using a systematic review design and thematic approach in the data analysis. The topic is of public health interest. The design is adequate and appropriate. The thoroughness in the execution of the project is commendable.

2. The Authors knowledge and use of existing literature on the subject:- The literature is deem to be adequate, but there is the need to have a definition of terms with respect to the determinants of the outcome of interest.

3. The specific contribution of the Dissertation to knowledge.

The study contribution of this research is in highlighting the special services provided for these age groups who might be marginalised to health care services by the facilities and its’ effect on the quality of services provided.

4. The literary style and technical presentation of the Dissertation.

The literary style is fluent and readable.

Final Statement:- The research work is recommended for publication as it is.

Reviewer #2: The definition of "older people" adopted for this study should be stated in this and all other resultant publications from this study. The UN definition of older people is people over 60 years but this study protocol demanded the recruitment of persons 60 years and above.

6. PLOS authors have the option to publish the peer review history of their article (what does this mean?). If published, this will include your full peer review and any attached files.

Reviewer #1: **Yes: **Adebayo T. Onajole

Reviewer #2: **Yes: **Udechukwu Felix Ezepue

---

## [Author Response · Author response to Decision Letter 0]

28 Dec 2023

1.Ensure that the manuscript meets PLOS ONE's style requirements: Done as per the Track Changes version

2. PubMed strategy: include number of articles at each stage: Done

3. Prisma flow diagram: add reasons for rejecting 930 articles in the first screening phase: The title and abstracts for 1004 records were screened and 930 were excluded as they did not meet the inclusion criteria (mentioned in the Prisma flow diagram). Specific reasons for excluding on title and abstract screening is not a requirement for Cochrane systematic reviews and is not a requirement in the Cochrane Handbook – current version 2023 MECIR Box 4.6.c Relevant expectations for conduct of intervention reviews. Title and abstract screening was completed by two reviewers working independently and a third reviewer was responsible for managing discrepancies before moving to full text review. 

4. Amendments to the protocol mentioned in the letter to editor should be reflected in the methods:In the letter we mentioned that the research question was narrowed down. The protocol included 5 sub-questions. We cited clearly that data presented in the manuscript report on two main questions related to experiences of older people with access, and access barriers and enablers. Summarized data answering the remaining questions mentioned in the protocol are available in the S2 file of the supporting information (manuscript lines 359-376). The removal of LILACS database is reported in line 380. A consultation with older people is reported in the protocol and was not included in the reporting of this systematic review. The rationale for this was the delay in writing the protocol to completing the review – and the Lebanon was impacted by economic downturn, Beirut blast and COVID 19. It was not possible to gain ethical approval and arrange to meet with older people during this challenging period. This component will be part of the interviews in phase 2 of the research and questions developed for interviews and focus groups will be informed by this systematic scoping review. 

5. Definition of terms with respect to the determinants of the outcome: Health care access is defined (line 137); primary health care is defined (line 74); older people defined (line 134).

The definition of "older people" adopted for this study should be stated in this and all other resultant publications from this study. The UN definition of older people is people over 60 years, but this study protocol demanded the recruitment of persons 60 years and above. Older people are defined in line 134 of the manuscript. According to UN published reports mentioned below, older people are those aged 60 or over. We adopted this definition in the protocol and selected relevant studies accordingly.

(a. United Nations Department of Economic and Social Affairs (UNDESA). World population aging; 2015. Page 1. Available from: https://www.un.org/en/development/desa/population/publications/pdf/ageing/WPA2015_Report.pdf

b. United Nations, Department of Economic and Social Affairs (UNDESA). Income Poverty in Old Age: an Emerging Development Priority [Internet]. Place unknown: UNDESA; 2019. Page 2. Available from: https://www.un.org/esa/socdev/ageing/documents/PovertyIssuePaperAgeing.pdf

c. United Nations Development Programme. Ageing, Older Persons, and the 2030 Agenda for sustainable development. (No date). Page 8. Available from: https://www.un.org/development/desa/dspd/wp-content/uploads/sites/22/2017/08/Ageing-Older-Persons-and-2030-Agenda_Issues-Brief-low-resolution-.pdf)

6. Review reference list: Done

7. Upload figure files to PACE digital diagnostic tool to ensure they meet PLOS requirements: Done

---

## [Decision Letter · Decision Letter 1]

2 Feb 2024

Access of older people to primary health care in low and middle-income countries: a systematic scoping review

PONE-D-23-19442R1

Dear Dr. Saydeh Dableh

We’re pleased to inform you that your manuscript has been judged scientifically suitable for publication and will be formally accepted for publication once it meets all outstanding technical requirements.

Kind regards,

Dirceu Henrique Paulo Mabunda, M.D.

Academic Editor

PLOS ONE

Additional Editor Comments (optional):

Reviewers' comments:

Reviewer's Responses to Questions

**Comments to the Author**

1. If the authors have adequately addressed your comments raised in a previous round of review and you feel that this manuscript is now acceptable for publication, you may indicate that here to bypass the “Comments to the Author” section, enter your conflict of interest statement in the “Confidential to Editor” section, and submit your "Accept" recommendation.

Reviewer #1: All comments have been addressed

Reviewer #2: All comments have been addressed

Reviewer #3: (No Response)

2. Is the manuscript technically sound, and do the data support the conclusions?

Reviewer #1: Yes

Reviewer #2: Yes

Reviewer #3: Yes

3. Has the statistical analysis been performed appropriately and rigorously? 

Reviewer #1: Yes

Reviewer #2: Yes

Reviewer #3: Yes

4. Have the authors made all data underlying the findings in their manuscript fully available?

Reviewer #1: Yes

Reviewer #2: Yes

Reviewer #3: Yes

5. Is the manuscript presented in an intelligible fashion and written in standard English?

Reviewer #1: Yes

Reviewer #2: Yes

Reviewer #3: Yes

6. Review Comments to the Author

Reviewer #1: The revised manuscript is deemed acceptable for publication.

The topic, ensuring access for older people to Primary Health Care (PHC) is vital to achieve

universal health coverage, improve health outcomes, and health-system performance. Older people living in Low-and Middle-Income Countries (LMICs) face barriers constraining their timely access to appropriate care. These are usually as a result of absence to socio-economic policies not targeting these age groups. Older people’s experiences with PHC access varied according to local contexts, socioeconomic variables, and the provision of public or private health services. Results inform policymakers and PHC practitioners to generate policies and services that are evidence-based and responsive to older people’s needs. Identified knowledge gaps highlight the need for research to further understand older people’s access to PHC in different LMICs.

Reviewer #2: (No Response)

Reviewer #3: Dear authors, I enjoyed reading your manuscript titled “Access of older people to primary health care in low and middle-income countries: a systematic scoping review”. It is indeed, well written, and flows quite well. You also followed a thorough process in your review and covered all the basics that need to be covered for a review.

I can also see that you have addressed the previous reviewers’ comments and I believe your article should be published. I have just a few things for you to consider which might strengthen this great paper a bit more.

1. Since no language filters were used, how did you account for translating the studies that are not in English (since you are publishing in a journal that publishes mainly in English)? For instance, for the articles in Portuguese, how did you translate these or are any of the authors Portuguese speakers?

It is important to include this information as we all know how meaning could be lost in the process of translation.

Also, were there other articles written in other languages (apart from English and Portuguese) that were identified from the search?

Might be helpful to include information on these to clear any ambiguities prior to publication.

2. You state that the results inform policymakers and PHC practitioners to generate policies and services that are evidence-based and responsive to older people’s needs.

Could be helpful if you could take this a bit further by stating how, giving a few examples or suggestions of policies and services that are based on evidence and could be helpful in responding to older people’s needs.

3. You also state that the identified knowledge gaps highlight the need for research to further understand older people’s access to PHC in different LMICs. While I can see you have made some general suggestions regarding these, it would be great if you could buttress this more by suggesting a research agenda or at least highlighting a couple of areas this research could focus on or research questions this could answer.

Again, these are just suggestions for you to consider to strengthen your paper and the great work you have put in towards the review.

7. PLOS authors have the option to publish the peer review history of their article (what does this mean?). If published, this will include your full peer review and any attached files.

Reviewer #1: **Yes: **ADEBAYO ONAJOLE

Reviewer #2: **Yes: **Udechukwu Felix Ezepue

Reviewer #3: No

---

## [Editor Report · Acceptance letter]

4 Mar 2024

PONE-D-23-19442R1 

PLOS ONE

Dear Dr. Dableh, 

I'm pleased to inform you that your manuscript has been deemed suitable for publication in PLOS ONE. Congratulations! Your manuscript is now being handed over to our production team.

Kind regards, 

on behalf of

Dr. Dirceu Henrique Paulo Mabunda 

Academic Editor

PLOS ONE